# Regulation of hepatic microRNAs in response to early stage *Echinococcus multilocularis* egg infection in C57BL/6 mice

Ghalia Boubaker[1,2]*, Sebastian Strempel[3], Andrew Hemphill[1], Norbert Müller[1], Junhua Wang[1], Bruno Gottstein[1,4]*, Markus Spiliotis[1]¤

1 Department of Infectious Diseases and Pathobiology, Institute of Parasitology, University of Bern, Bern, Switzerland, 2 Department of Clinical Biology B, Laboratory of Parasitology and Mycology, University of Monastir, Monastir, Tunisia, 3 Microsynth AG, Balgach, Switzerland, 4 Institute of Infectious Diseases, Faculty of Medicine, University of Berne, Berne, Switzerland

¤ Current address: University of Würzburg, Institute of Hygiene and Microbiology, Würzburg, Germany
* ghalia.boubaker@vetsuisse.unibe.ch (GB); bruno.gottstein@ifik.unibe.ch (BG)

## Abstract

We present a comprehensive analysis of the hepatic miRNA transcriptome at one month post-infection of experimental primary alveolar echinococcosis (AE), a parasitic infection caused upon ingestion of *E. multilocularis* eggs. Liver tissues were collected from infected and non-infected C57BL/6 mice, then small RNA libraries were prepared for next-generation sequencing (NGS). We conducted a Stem-loop RT-qPCR for validation of most dysregulated miRNAs. In infected mice, the expression levels of 28 miRNAs were significantly altered. Of these, 9 were up-regulated (fold change (FC) $\geq$ 1.5) and 19 were down-regulated (FC $\leq$ 0.66) as compared to the non-infected controls. In infected livers, mmu-miR-148a-3p and mmu-miR-101b-3p were 8- and 6-fold down-regulated, respectively, and the expression of mmu-miR-22-3p was reduced by 50%, compared to non-infected liver tissue. Conversely, significantly higher hepatic levels were noted for *Mus musculus* (mmu)-miR-21a-5p (FC = 2.3) and mmu-miR-122-5p (FC = 1.8). In addition, the relative mRNA expression levels of five genes (*vegfa*, *mtor*, *hif1-α*, *fasn* and *acsl1*) that were identified as targets of down-regulated miRNAs were significantly enhanced. All the five genes exhibited a higher expression level in livers of *E. multilocularis* infected mice compared to non-infected mice. Finally, we studied the issue related to functionally mature arm selection preference (5p and/or 3p) from the miRNA precursor and showed that 9 pre-miRNAs exhibited different arm selection preferences in normal versus infected liver tissues. In conclusion, this study provides first evidence that miRNAs are regulated early in primary murine AE. Our findings raise intriguing questions such as (i) how *E. multilocularis* affects hepatic miRNA expression;(ii) what are the alterations in miRNA expression patterns in more advanced AE-stages; and (iii) which hepatic cellular, metabolic and/or immunologic processes are modulated through altered miRNAs in AE. Thus, further research on the regulation of miRNAs during AE is needed, since miRNAs constitute an attractive potential option for development of novel therapeutic approaches against AE.

**Data Availability Statement:** All relevant data are within the manuscript and its Supporting Information files.

**Funding:** This work was financially supported by the Gottfried und Julia Bangerter-Rhyner-Stiftung (MS and GB) [http://www.bangerter-stiftung.ch/bangerter/de/stiftung/portrait.html] and by the Swiss National Science Foundation (BG, AH) (grant no. 31003A_160108/1) [http://www.snf.ch/en/Pages/default.aspx]. The funders had no role in study design, data collection and analysis, decision to publish, or preparation of the manuscript.

**Competing interests:** I have read the journal's policy and the authors of this manuscript have the following competing interests: Sebastian Strempel is employee of Microsynth AG.

## Author summary

Various infectious diseases in humans have been associated with altered expression patterns of microRNAs (miRNAs), a class of small non-coding RNAs involved in negative regulation of gene expression. Herein, we revealed that significant alteration of miRNAs expression occurred in murine liver subsequently to experimental infection with *E. multilocularis* eggs when compared to non-infected controls. At the early stage of murine AE, hepatic miRNAs were mainly down-regulated. Respective target genes of the most extensively down-regulated miRNAs were involved in angiogenesis and fatty acid synthesis. Furthermore, we found higher mRNA levels of three angiogenic and two lipogenic genes in *E. multilocularis* infected livers compared to non-infected controls. Angiogenesis and fatty acid biosynthesis may be beneficial for development of the *E. multilocularis* metacestodes. In fact the formation of new blood vessels in the periparasitic area may ensure that parasites are supplied with oxygen and nutrients and get rid of waste products. Additionally, *E. multilocularis* is not able to undertake de novo fatty acid synthesis, thus lipids must be scavenged from its host. More research on the regulation of the hepatic miRNA transcriptome at more advanced stages of AE is needed.

## Introduction

Human alveolar echinococcosis (AE) is a parasitic disease caused by infection with the larval stage (metacestode) of the cyclophyllidean tapeworm *E. multilocularis* (Cestoda, Taeniidae) [1]. In Europe, *E. multilocularis* undergoes a sylvatic life cycle that predominantly includes the red fox (*Vulpes vulpes*) as major definitive host (HD) and rodents (family Arvicolidae) acting as intermediate hosts (IHs) [2]. For humans, accidental infection with *E. multilocularis* eggs through the oral route can lead to the development of AE, affecting primarily the liver in 98% of cases [1]. Within the liver tissue, the asexual proliferation of metacestodes occurs by exogenous budding of new vesicles, thus the larval mass progrediently invades the surrounding hepatic tissue, with a potential of metastasis formation in distant sites such as lungs, brain and other organs [3–6]. Disease progression is largely supported by the ability of *E. multilocularis* metacestodes to modulate immunological host-defense mechanisms [7,8].

AE is listed as one of the rare and neglected tropical diseases by the World Health Organization, and if left untreated the disease results in mortality in more than 90% of cases [9]. Radical surgical excision of the parasite tissue complemented by adjuvant chemotherapy is the only curative treatment for hepatic AE, but this applies to only ~ 30% of patients [10]. For inoperable cases, the only currently licensed chemotherapeutical option is based on the benzimidazole derivatives albendazole and mebendazole. Benzimidazole-therapy has increased the survival rate of affected patients to 85–90% [4]. However, these drugs exhibit a parasitostatic rather than parasitocidal activity; hence patients must take these drugs lifelong. Currently, there is no alternative to benzimidazole-based chemotherapy for patients suffering from benzimidazole intolerance [11]. The claim for a better management and control of AE calls for new treatment concepts, thus highlighting once more the necessity to gain deeper insights into the molecular basis of AE-induced liver pathology.

MicroRNAs (miRNAs) are a class of 21–24 nucleotides (nt) small non-coding RNAs discovered in the early 1990s as key regulatory factors of developmental timing in *Caenorhabditis elegans* [12]. The biogenesis of miRNAs is a two-step process; it begins in the nucleus where a

miRNA gene is transcribed to primary miRNA (pri-miRNA) which will be processed by nuclear RNase III Drosha-like nucleases to generate a precursor hairpin miRNA (pre-miRNA). This pre-miRNA is then exported to the cytoplasm to become a mature miRNA [13,14]. A miRNA can specifically bind to its target mRNA at the 3' untranslated regions (3'UTRs), 5' UTRs, exons and/or introns [15,16]. This results in repression of the target mRNA expression by diverse mechanisms, including inhibition of the translational machinery, disruption of cap–poly (A) tail interactions, and exonuclease-mediated mRNA cleavage [17]. Thus, miRNAs are major elements of negative post-transcriptional regulation of gene expression. The numbers of human miRNA-generating loci are continuously increasing and range between 2000 to 4000 [18–21]. Based on computational predictions it is estimated that more than 60% of all human protein-coding genes harbor at least one conserved miRNA-binding site [22,23]. Several cellular and biological processes such as cell proliferation, metabolism, apoptosis and immune defenses are orchestrated by miRNAs [24–26].

Alterations in microRNA gene expression have been reported for a wide range of human pathologies such as cancers, metabolic disorders and cardiovascular diseases [27,28]. In hepatocellular carcinoma (HCC), dysregulated miRNAs have been assessed as drug targets [29,30], disease stage / survival rate predictors [31] and as markers of responsiveness to therapy [32].

In recent years, significant efforts have been made in outlining and defining the roles of miRNAs in host-pathogen interactions, with a main focus on host miRNAs. In this context, evidence was provided that hepatitis C virus replication is completely dependent on the liver-specific miR-122 [33]. Similarly, changes of host-miRNA expression profiles have been associated to different helminthiases [34–37] where miRNAs dysregulation was relevant to tissue dysfunction and to the type of immune response [38]. In the case of *Schistosoma japonicum*, high serum level of hepatic miR-223 was correlated with active infection, and responsiveness to praziquantel therapy was characterized by a return to normal levels [39]. *E. multilocularis* was also found to quantitatively modulate circulating and liver miRNAs in a mouse model of secondary (intraperitoneal) infection [40,41]. Similarly, the closely related *E. granulosus* also caused changes in the intestinal miRNA transcriptome of Kazakh sheep [42].

To date, miRNA-directed therapy against helminthiases is highly appealing [43]; both parasite—and host- derived miRNAs can be targeted, which allows (i) to interfere in essential biological processes of the parasite [44–46] and (ii) to ensure that the host environment returns to a normal biological state [47].

Growth of *E. multiocularis* larvae induces changes to liver metabolism that collectively result in a net mobilization of glucose, lipid and amino acids [48,49]. Other studies demonstrated deviations in hepatic gene expression at early stage of experimental primary *E. multilocularis* infection in the murine model compared to non-infected liver tissue [50,51]; however, how and whether miRNAs are involved in post-transcriptional regulation of gene expression remains unknown. In immunocompetent individuals, human AE usually takes decades before symptoms arise, and once diagnosed, patients often have already reached an advanced stage of disease hampering the prospect to achieve complete parasite clearance. Accordingly, understanding of early cellular and molecular events that take place along with intra-hepatic establishment of *E. multilocularis* larvae is a crucial task for the development of novel means of prevention and treatment.

In this study, we applied Illumina next-generation sequencing (NGS) to comprehensively analyze the miRNAs expression profile in the mouse liver at the early stage (one month post-infection) of primary AE, and compared miRNAs expression levels in infected livers to non-infected tissue samples. Results were validated based on quantitative stem-loop RT-PCR. Hepatic miRNAs that exhibited significantly altered disease specific expression levels were further studied by Reactome and KEGG enrichment analyses. Furthermore, we used infected and

non-infected livers tissue samples to comparatively measure the relative mRNA levels of five genes that have been identified as targets of dysregulated miRNAs. This is the first survey of miRNAs regulation in early AE, which will contribute to a better understanding of the role of hepatic miRNAs in promoting parasite survival and growth.

## Materials and methods

### Ethics statement

Mice were housed and handled under standard laboratory conditions and in agreement with the Swiss Animal Welfare Legislation (animal experimentation license BE 103/11 and BE 112/14).

### Mouse model and primary AE infection

Ten 8-week-old female *C57BL/6* mice were obtained from Charles River GmbH, Germany. The animals were divided into two groups, five animals each. The control group remained uninfected, and the other group was orally infected with *E. multilocularis* eggs. The eggs used in this study were obtained from a naturally infected fox that was shot during the official Swiss hunting season. Parasite eggs for subsequent infection of mice were prepared as described by Deplazes & Eckert (1996) [52]. Briefly, the fox intestine was removed under appropriate safety precautions and cut into 4 pieces. After longitudinal opening of the intestinal segments, the worm-containing mucus was scraped out and put into petri dishes containing sterile water. Subsequently, the mucosal suspension was serially filtered through a 500μm and then 250μm metal sieve, by concurrently disrupting the worms with an inversed 2 ml syringe top. This suspension was further filtered through a 105 μm nylon sieve. The eggs were then washed by repeated sedimentation (1xg, 30 min., room temperature) in sterile water containing 1% Penicillin/Streptomycin and stored in the same solution at 4 ˚C. The sodium hypochlorite resistance test was used to assess egg viability [53], and to ascertain efficiency of infectivity, a preliminary test was done in two female *C57BL/6* mice. After confirmation of infectivity, five mice were each inoculated with approximately $1 \times 10^3$ eggs suspended in 100 μl sterile water by gavage. The control mice received sterile water only. At one month post-infection, all animals were euthanized, and their livers were removed under sterile conditions for further analyses. All animal infections were performed in a biosafety level 3 unit (governmental permit no. A990006/3). One mouse from the uninfected control group died during the experiment from unknown causes and was thus not included in this experiment.

### Liver samples, total RNA extraction, small RNA library preparation and NGS

Liver tissue samples from *E. multilocularis*-infected mice were taken from the peri-parasitic area, precisely 3 to 4 mm adjacent to parasite lesions, which appeared as small white or yellowish spots. Liver tissue samples from uninfected animals were collected from the same hepatic areas. The obtained liver tissues were minced quickly, mixed at a ratio of 1:10 with QIAzol lysis reagent (Qiagen, Cat: 79306) and homogenized by bead beating (FastPrep-24 Instrument, MP Biomedicals, Cat: 116004500). Total RNA was extracted according to the manufacturer's instruction of the QIAzol lysis reagent with the exception that chloroform was replaced by 1-bromo-3-chloropropane for the phase separation step (Sigma, cat: B9673). To remove genomic DNA contamination in RNA samples, an enzymatic digestion step using DNase I (Thermo Fisher Scientific, Cat: EN0521) was carried out. Finally, total RNA was re-suspended

in RNase-free water. RNA quantity and RNA quality number (RQN ≥ 8) was determined by the Fragment Analyzer CE12 (Advanced Analytics).

Two to five RNA extractions were prepared simultaneously from each liver, in order to subsequently choose the preparation with the best RQN. Samples from the infected mouse group were named as follows; AE-1pm-1.1, AE-1pm-2.1, AE-1pm-3.2, AE-1pm-4.1 and AE-1pm-5.1. Total RNA preparations from the uninfected control group were labelled ctr-1pm-1.1, ctr-1pm-3.2, ctr-1pm-4.2 and ctr-1pm-5.1 (first digit is the number of the mouse within its corresponding group and the second digit indicates the number of RNA preparation).

Sequencing library construction and sequencing itself were performed by Microsynth (Balgach, Switzerland). Briefly, five libraries were generated from five mice (three mice from *E. multilocularis*-infected group and two mice from the uninfected control group). The total RNA was checked on an Agilent 2100 Bioanalyzer instrument for degradation. Subsequently, the CleanTag Ligation Kit (TriLink BioTechnologies) was used to prepare small RNA stranded libraries from total RNA (1µg RNA per library). Libraries were analyzed a second time on the Bioanalyzer to check for the expected miRNAs fragment peak at 141 bp and subsequently a Sage Science Pippin Prep instrument was used to select the expected fragment size range. In a last step before the deep sequencing, the quality and concentration of each final library (the eluted 141 bp band) were submitted to PicoGreen analysis (Thermo Fisher Scientific, Cat: P7589). The so refined libraries were then sequenced on an Illumina NextSeq 500 instrument using a high-output v2 Kit (75-cycles sequencing run) and targeting for an output of 50 million pass-filter reads.

Raw sequence data, processed- and metadata generated in this study have been deposited in NCBI Gene Expression Omnibus (GEO) repository (http://www.ncbi.nlm.nih.gov/geo/) under the following accessions; GSM4367926 (ctr-1pm-3.2), GSM4367927 (ctr-1pm-4.2), GSM4367928 (AE-1pm-2.1), GSM4367929 (AE-1pm-3.2) and GSM4367930 (AE-1pm-5.1).

## Bioinformatics analysis of small RNA sequencing data

In a preprocessing step, reads were subjected to de-multiplexing and trimming of the TriLink adapter residuals using Illumina bcl2fastq v2 analysis software (bcl2fastq2 Conversion Software v2.19.1.). Quality of reads was checked with the software FastQC (v. 0.11.5) (https://www.bioinformatics.babraham.ac.uk/projects/fastqc/); thus, reads shorter than 10 bases or longer than 25 bases, or reads containing any "N" base, were discarded to refine the input for the statistical analysis. In a next step, clean reads of samples derived from the same experimental group (AE-1pm or ctr-1pm) were pooled together and then dereplicated using the software usearch (v. 8.1.1681) [54]. This resulted in a list of unique sequences that were annotated and recorded based on their frequency of occurrence. Within each dataset (AE-1pm and ctr-1pm), sequences having a read count of at least 10 were blasted against the miRBase [55,56] mature miRNAs mouse sequences (*Mus musculus*, mouse mm10 reference genome). Mapping of cleaned reads to the mouse genome (v. mm10) was performed using the RNA mapping software STAR (v. 2.5.1b) [57]. Major parameters used were a minimum of 16 matching bases between the read and the reference, whereas a maximum of 1 mismatch was allowed and splice aware mapping was switched off. Raw counting of uniquely mapped reads to annotated mature miRNAs was done with the software htseq-count from the HTSeq software suite (v. 0.6.1p1) [58]. The miRNAs annotations stem from miRBase matching the mm10 reference genome.

Lastly and in an exploratory step to check for what else might be present in our data in addition to what the miRBase provided, we performed a blast against the Rfam collection of RNA families (v. 12.3) [59]. This blast concerned only sequences that did not show a hit with an e-

value of at least 1e-6 or smaller during the blast against the miRBase. Therefore for the blast against Rfam, we choose an Expect value (E) threshold of less than 1; the E- value criteria describe the number of expected hits occurring by chance. According to this, we considered hits with an E-value ranged between 0 and 1 as significant matches.

Normalization of the raw miRNAs read counts and differential miRNAs expression analysis were conducted using the R package DESeq2 (v. 1.12.4); the DESeq2 method combines shrinkage estimation for dispersions and fold changes (log2FC) which allow a more sophisticated quantitative analysis [60,61].

We applied principal component analysis in two dimensions (2D-PCA) [62] to check whether control mice (ctr-1pm) could be distinguished based on their miRNAs expression profiles from those experimentally infected with *E. multilocularis* eggs. The PCA statistical approach is applied to complex data sets and aims to reduce the dimensionality of the data matrix while simultaneously retaining the maximum amount of variance. The term 2D refers to the number of used principal components: the first component, PC 1, represents the direction of the highest variance of the data and PC 2 points into the directions of the highest of the remaining variance orthogonal to the first component.

In parallel to 2D-PCA, we also applied the hierarchical clustering method [63] for analyzing hepatic miRNA transcriptome data from infected and non-infected mice. Hierarchical clustering consists in building clusters of libraries with similar patterns of expression. Results are displayed as a tree diagram (dendrogram).

## Stem-loop reverse transcription (RT) and real-time quantitative (Stem-loop RT-qPCR) of dysregulated mature miRNAs

To validate the expression profile of the most dysregulated miRNAs obtained from the analysis of NGS data, we assessed the relative expression levels of twelve miRNAs by stem-loop RT-qPCR, as previously described [64]. The small nucleolar RNA 234 gene (*sno234*) was included as reference for microRNA normalization [65]. In total, 9 liver tissue samples derived from the AE-infected (5 mice) and uninfected mice (4 mice) were individually assessed.

Briefly, 1 µg of total RNA template was reverse transcribed to cDNA by M-MLV reverse transcriptase (Promega) using specific stem-loop RT primer and under the following conditions: 5 min at 25 °C, 60 min at 42°C, 15 minutes at 70°C, reactions were then cooled down to 4°C. All q-PCRs were performed in Rotor-Gene 6000 (Corbett Life Science) and using FastStart Essential DNA Green Master Kit (Roche, Switzerland). The total volume for qPCR was 10 µl, consisting of 5 µl of Faststart Essential DNA Green Master Mix (2xconc), 1 µl of forward and reverse primers, and 2.5 µl of cDNA template (diluted 1:5). All experiments were run in triplicates. Quantitative PCRs were performed according to the following program: an initial hold at 95°C for 15 min, 40 cycles (94°C for 15 s, annealing at 63°C for 20 s, extension at 72°C for 20 s) and a final denaturation step from 50 °C to 95 °C. Specificity of each qPCR and presence of primer dimers were checked based on analysis of the generated melting curve. All primers characteristics are listed in S1 Table.

For relative miRNA expression, data were expressed as median ± standard deviation (SD) and examined for statistical significance with the nonparametric Mann–Whitney U test. *P*-values of less than 0.05 were considered to be statistically significant.

## MicroRNA target prediction and pathway enrichment analysis

Significantly differentially expressed miRNAs with FC $\geq$ 1.5 or FC $\leq$ 0.66 were chosen for target prediction. Since miRNAs inhibit their target mRNAs, reduction of miRNA expression lead to up-regulation of the target genes and vice versa. For miRNA-mRNA target predictions,

we used miRNet [66], a database for network-based visual analysis of miRNAs, targets and functions. MiRNet integrates high-quality miRNA-target interaction databases (miRTarBase v6.0, TarBase v6.0 and miRecords); these databases provide direct experimental evidence regarding the miRNA–target interaction. For functional and pathway enrichment analysis, we used two pathway databases, including Reactome [67] and KEGG [68]. Statistical significance (P < 0.01) was measured by applying hypergeometric test [66].

Furthemore, miRNet allowed us to visualize miRNA-target interactions in a network context.

Thus, pathway enrichment analysis and visualization of interaction networks can be reproduced at any time using miRNet, a web-based tool freely available at http://www.mirnet.ca. For that, the list of down- or up- regulated miRNAs (miRBase Accession) is used as input; selected parameters are *M. musculus* (mouse) for organism, miRBase Accession for the ID type and genes for target type. When choosing KEGG or Reactome database, a table containing the list of pathways and the number of involved genes will be shown. By clicking on a given pathway, the names of involved genes will be displayed.

### Expression analysis of miRNA target genes using RT-qPCR

We used RT-qPCR to comparatively assess the relative expression of five genes involved in angiogenesis and fatty acid biosynthesis and activation. The three pro-angiogenic genes were vascular endothelial growth factor A (VEGFA), the mechanistic target of rapamycin (MTOR); and the hypoxia inducible factor 1, $\alpha$ subunit (HIF1$\alpha$). The two examined lipogenic genes were fatty acid synthase (FASN) and acyl-CoA synthetase long-chain family member 1 (ACSL1). These genes were chosen using the following criteria: (1) predicted as a target of at least one of the down-regulated miRNA; (2) experimentally validated target of at least one of the down-regulated miRNA; (3) significantly relevant gene in the considered pathway; (4) a combination of 1, 2 and 3 (S2 Table).

All genes were assessed individually in five liver tissue samples from AE-infected and four samples from uninfected mice (same experiment). Thus, nine cDNA preparations were synthesized on the same RNA templates which were used for validation of most dysregulated miRNAs by Stem-loop RT-qPCR (see section above). The qPCRs were performed as described above, with the exception that the annealing temperature was set at 62˚ C for all five genes. We used the glyceraldehyde-3-phosphate dehydrogenase (*gapdh*) as endogenous reference. Detailed information on qPCR primers are provided in S2 Table.

## Results

### Next generation sequencing (NGS) data

To characterize the miRNA transcriptome in murine liver during early stage of primary AE, small RNA libraries from three *E. multilocularis*-infected and two uninfected control mice were constructed and subjected to high-throughput sequencing.

The number of uniquely mapped reads with an average mapped length of 21 nucleotides ranged from 4.831.114 to 1.749.597, representing thus 66 to 60% of the total cleaned reads. More than 95% of those uniquely mapped reads were assigned to miRBase annotated miRNAs (Table 1).

For all libraries, length distribution analyses revealed that length of most abundant sequences ranged from 18 to 24 nt with a peak at 21 nt (S1A Fig). In both groups, most miRNAs were detected with read counts below 100 (S1B Fig).

**Table 1. Summary of counts for clean reads, unique mapped reads and reads on feature of miRNAs.**

| Library | [a]Cleaned Reads | [b]Uniquely Mapped Reads | [c]Reads On Feature | Reads No Feature | Reads Ambiguous |
|---|---|---|---|---|---|
| AE-1pm-2.1 | 6802673 | 3840087 | 3659366 | 180718 | 3 |
| | | | 95.3% | 4.7% | 0% |
| AE-1pm-3.2 | 7315799 | 4831114 | 4698825 | 132287 | 2 |
| | | | 97.3% | 2.7% | 0% |
| AE-1pm-5.1 | 7098967 | 4591014 | 4459799 | 131210 | 5 |
| | | | 97.1% | 2.9% | 0% |
| ctr-1pm-3.2 | 2901675 | 1749597 | 1690171 | 59426 | 0 |
| | | | 96.6% | 3.4% | 0% |
| ctr-1pm-4.2 | 6683305 | 3861026 | 3704668 | 156358 | 0 |
| | | | 96% | 4% | 0% |

[a]: reads considered for mapping

[b]: the reads that mapped to exactly one location within the reference genome

[c]: features of miRNAs

## Overview of miRNA expression profile in infected and uninfected mouse liver

A total of 699 known miRNAs were identified from both infected (AE-1pm) and uninfected (ctr-1pm) groups. Among these molecules, 530 were common to both groups with 124 and 45 miRNAs specifically expressed in the AE-1pm and ctr-1pm libraries, respectively. Specific miRNAs to AE-1pm (124 miRNAs) or to ctr-1pm (45 miRNAs) libraries were all present with read counts below 100. From the co-expressed miRNAs cluster, a set of 87 miRNAs with a read count > 1000 in at least one of the two experimental groups was identified (Fig 1A). In liver-tissue samples derived from AE-infected mice, mmu-miR-122-5p, mmu-miR-21a-5p and mmu-miR-192-5p accounted for up than 70% of the total normalized miRNAs counts, and they still comprised more than 50% total miRNAs abundance in the control group (Fig 1B).

## Alteration of the hepatic microRNA expression profile at early stage of hepatic AE

The 2D-PCA (Fig 2A) as well as the heatmap of sample-to-sample distances (Fig 2B) showed a clear separation of miRNA expression patterns between *E. multilocularis* egg infection versus healthy control. Globally, the observed clustering of samples from the same group indicated that a change in miRNAs expression pattern had occurred following infection.

A global comparative analysis of all miRNA read counts between both experimental mouse groups was carried-out and revealed the presence of a set of miRNAs whose fold change was significant. From this latter cluster, we considered as significantly dysregulated miRNAs only those with (i) a normalized read count $\geq$ 1000, and (ii) FC $\geq$ 1.5 (Log2FC $\geq$ 0.58) or FC $\leq$ 0.66 (Log2FC $\leq$ -0.58). Thus, a total of 28 miRNAs were found to be differentially expressed in diseased livers compared to healthy controls (Fig 3).

More information on detailed counts is shown in Table 2. The highest up-regulated miRNA in *E. multilocularis* infected livers was mmu-miR-21a-5p with a FC = 2.3. Conversely, the expression of mmu-miR-148a-3p was ~8-fold lower as compared to control liver samples.

Stem-loop RT-qPCR was applied to validate the miRNAs NGS data of 12 out of the 28 differentially expressed miRNAs. In order to optimize the statistical significance of this study,

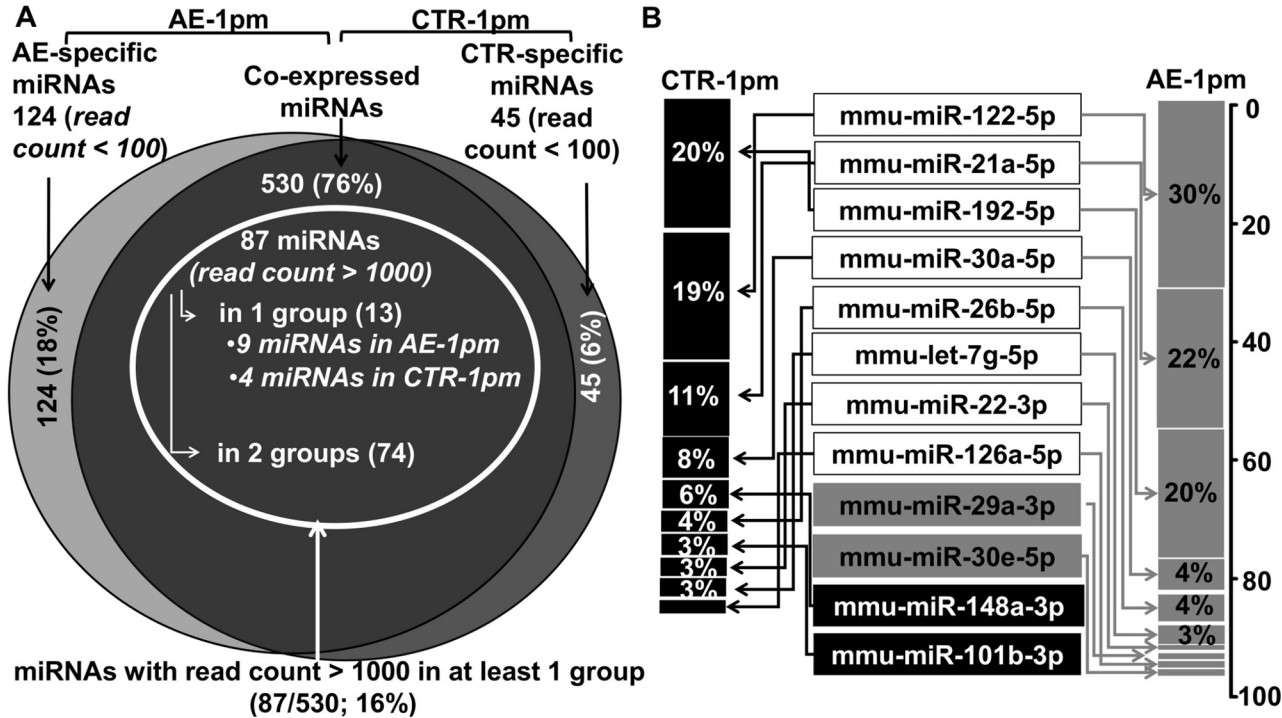

**Fig 1. miRNA expression profile in livers of AE-infected and uninfected mice. (A)** Venn diagram showing the overlap of expressed miRNA in liver-tissue samples from AE- infected and uninfected animals. **(B)** Top 10 most abundant miRNA and their frequency (%) in both groups. The two miRNAs in gray rectangles were present only among the top 10 expressed hepatic miRNAs in AE- infected mice, whereas miRNAs in black rectangles were only among the top 10 expressed hepatic miRNAs in uninfected control mice.

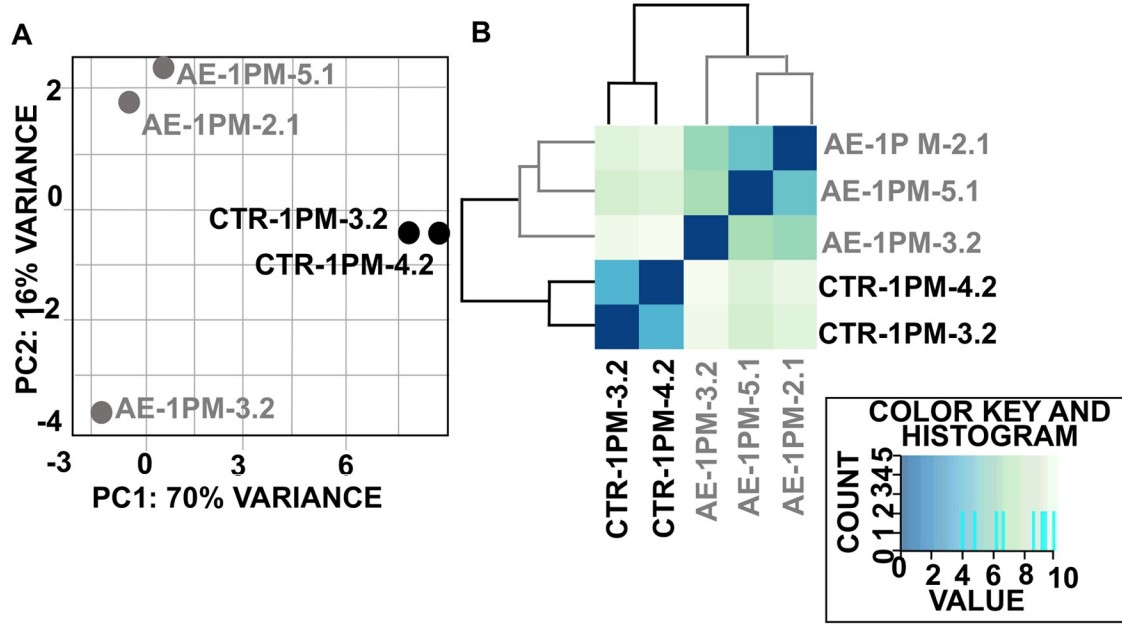

**Fig 2. Principal component analysis (A) and (B) hierarchical clustering both revealed a clear separation between hepatic miRNA profiles from infected and uninfected mice.** All the 699 miRNAs, identified in this study, with the normalized read-count in each animal group, are listed in S3 Table.

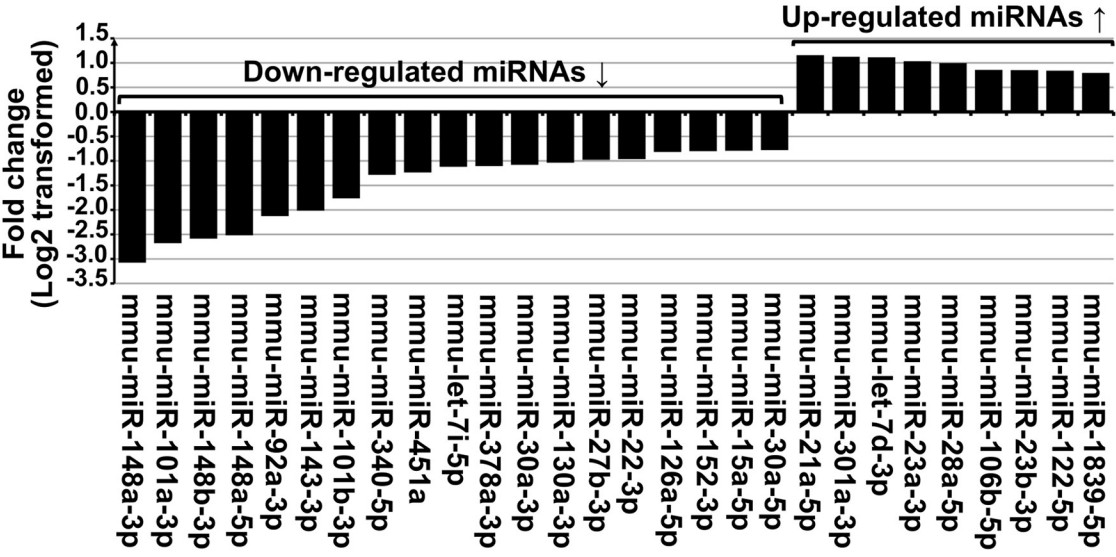

**Fig 3. Twenty-eight dysregulated hepatic miRNAs in primary AE-infected mice compared to control healthy individuals: Nine were up-regulated (Log2FC $\geq$ 0.58) and nineteen were down-regulated (Log2FC $\leq$ -0.58).**

two additional samples were included into each group, thus the AE-1pm group comprised five animals and the ctr-1pm included four samples. Overall, the Stem-loop RT-qPCR results largely confirmed the results obtained through NGS as shown in Fig 4 (Fig 4). Seven miRNAs; mmu-miR-148a-3p, mmu-miR-143-3p, mmu-miR-101b-3p, mmu-miR-340-5p, mmu-miR-22-3p, mmu-miR-152-3p, mmu-miR-30a-5p were significantly less expressed in AE-1pm samples compared to ctr-1pm samples. In contrast, infected mice exhibited significantly higher expression levels of mmu-miR-21a-5p, mmu-miR-28a-5p, mmu-miR-122-5p and mmu-miR-1839-5p compared to the control mice. No significant difference in expression levels in the two groups were noted for mmu-miR-301a-3p.

## miRNA precursors: Preference selection of 5p- and 3p arm in *E. multilocularis*-infected–and uninfected liver tissues

In order to interrogate arm selection preferences of miRNA pairs (miRNA-5p/3p) in normal and infected liver tissues, we investigated the expression levels of miR-5p and miR-3p strands of the same miRNA precursor (pre-miRNA). We identified a total of 127 miRNAs pairs that were co-expressed in all tissue samples and that exhibited a normalized read count $\geq$ 100 for 5p or 3p strand under infected and non-infected conditions. For each miRNA duplex, we calculated the selection rate (S) of 5p and 3p arm from the total read count (5p and 3p). For 25 (20%) and 21 (17%) out of these 127 miRNA pairs, mature miRNA was strictly derived from 3p (S%_miR-5p = 0; S%_miR-3p = 100) and 5p (S%_miR-5p = 100; S%_miR-3p = 0), respectively, and this was independent from the primary AE infection. Regarding the remaining 81 pre-miRNAs out of the total 127 identified pairs, mature miRNAs were derived from both arms. For 16 pre-miRNAs (out of the 81), 5p and 3p selection rates remained unchanged between infected and non-infected liver tissue, whereas selection preference for either the 5p- or the 3p-strand differed for the lasting 65 miRNA pairs (out of the 81). Linear regression analysis of 5p-arm selection rate in the 127 miRNA pairs showed a strong correlation (R2 = 0.9764) between the two groups of mice (Fig 5A). Particularly for nine pre-miRNA

**Table 2. Dysregulated miRNAs in primary AE.**

| miR_ID | Mean AE-1pm | AE-1pm-2.1 | AE-1pm-3.2 | AE-1pm-5.1 | Mean ctr-1pm-1pm | ctr-1pm-3.2 | ctr-1pm-4.2 | log2FC | p-Value | Adjusted p-Value | FC | % exp. |
|---|---|---|---|---|---|---|---|---|---|---|---|---|
| **Down-regulated** | | | | | | | | | | | | |
| mmu-miR-148a-3p | 21984 | 18178 | 20035 | 27738 | 185456 | 171132 | 199781 | -3.08 | 5.47E-41 | 2.84E-38 | 0.1 | 12 |
| mmu-miR-101a-3p | 8150 | 6877 | 7338 | 10235 | 52241 | 53256 | 51226 | -2.68 | 1.82E-33 | 3.15E-31 | 0.2 | 16 |
| mmu-miR-148b-3p | 1383 | 1264 | 1317 | 1568 | 8295 | 7122 | 9468 | -2.58 | 4.87E-34 | 1.27E-31 | 0.2 | 17 |
| mmu-miR-148a-5p | 209 | 173 | 228 | 227 | 1200 | 1083 | 1317 | -2.52 | 2.28E-28 | 2.96E-26 | 0.2 | 17 |
| mmu-miR-92a-3p | 608 | 633 | 454 | 737 | 2654 | 2409 | 2899 | -2.13 | 3.21E-18 | 2.62E-16 | 0.2 | 23 |
| mmu-miR-143-3p | 4046 | 3595 | 3678 | 4866 | 16340 | 15686 | 16994 | -2.01 | 3.55E-22 | 3.69E-20 | 0.2 | 25 |
| mmu-miR-101b-3p | 30534 | 25949 | 24758 | 40895 | 103675 | 103319 | 104031 | -1.76 | 9.17E-13 | 4.34E-11 | 0.3 | 29 |
| mmu-miR-340-5p | 2483 | 2019 | 2566 | 2863 | 6041 | 6087 | 5994 | -1.28 | 9.80E-10 | 3.18E-08 | 0.4 | 41 |
| mmu-miR-451a | 2403 | 2362 | 2968 | 1879 | 5655 | 5375 | 5935 | -1.23 | 8.31E-08 | 2.15E-06 | 0.4 | 42 |
| mmu-let-7i-5p | 10485 | 10057 | 9827 | 11571 | 22775 | 22279 | 23271 | -1.12 | 6.41E-10 | 2.38E-08 | 0.5 | 46 |
| mmu-miR-378a-3p | 11779 | 13061 | 7838 | 14438 | 25364 | 23747 | 26980 | -1.11 | 2.51E-05 | 2.90E-04 | 0.5 | 46 |
| mmu-miR-30a-3p | 6101 | 5610 | 4844 | 7850 | 12907 | 11685 | 14130 | -1.08 | 7.86E-06 | 1.24E-04 | 0.5 | 47 |
| mmu-miR-130a-3p | 6004 | 6156 | 6014 | 5843 | 12310 | 12379 | 12241 | -1.04 | 1.12E-09 | 3.42E-08 | 0.5 | 49 |
| mmu-miR-27b-3p | 19530 | 18479 | 17438 | 22672 | 38497 | 37696 | 39299 | -0.98 | 5.34E-07 | 1.16E-05 | 0.5 | 51 |
| mmu-miR-22-3p | 44461 | 48290 | 34592 | 50501 | 86810 | 87825 | 85795 | -0.97 | 8.37E-06 | 1.28E-04 | 0.5 | 51 |
| mmu-miR-126a-5p | 41918 | 38304 | 47589 | 39862 | 73898 | 74698 | 73098 | -0.82 | 1.30E-05 | 1.77E-04 | 0.6 | 57 |
| mmu-miR-152-3p | 5466 | 4954 | 5722 | 5722 | 9524 | 9966 | 9081 | -0.80 | 1.21E-05 | 1.74E-04 | 0.6 | 57 |
| mmu-miR-15a-5p | 2876 | 2888 | 2736 | 3004 | 4988 | 5104 | 4871 | -0.79 | 6.29E-06 | 1.02E-04 | 0.6 | 58 |
| mmu-miR-30a-5p | 147634 | 129644 | 125683 | 187576 | 253258 | 228418 | 278098 | -0.78 | 8.39E-04 | 6.61E-03 | 0.6 | 58 |
| **Up-regulated** | | | | | | | | | | | | |
| mmu-miR-21a-5p | 783632 | 556970 | 1144987 | 648938 | 351395 | 348508 | 354282 | 1.16 | 5.63E-05 | 6.10E-04 | 2.3 | 223 |
| mmu-miR-301a-3p | 1514 | 1336 | 1894 | 1312 | 694 | 730 | 657 | 1.13 | 1.29E-06 | 2.49E-05 | 2.2 | 218 |
| mmu-let-7d-3p | 1214 | 1326 | 1086 | 1230 | 561 | 585 | 538 | 1.11 | 3.33E-08 | 9.62E-07 | 2.2 | 216 |
| mmu-miR-23a-3p | 2728 | 2617 | 3302 | 2264 | 1330 | 1282 | 1379 | 1.04 | 2.56E-06 | 4.44E-05 | 2.1 | 205 |
| mmu-miR-28a-5p | 4292 | 5070 | 3932 | 3875 | 2152 | 2223 | 2081 | 1.00 | 1.11E-06 | 2.21E-05 | 2 | 199 |
| mmu-miR-106b-5p | 1620 | 1735 | 1699 | 1425 | 895 | 967 | 822 | 0.86 | 2.43E-05 | 2.87E-04 | 1.8 | 181 |
| mmu-miR-23b-3p | 5183 | 5412 | 5151 | 4986 | 2873 | 2897 | 2850 | 0.85 | 1.04E-06 | 2.16E-05 | 1.8 | 180 |
| mmu-miR-122-5p | 1081837 | 1071557 | 1177605 | 996349 | 603346 | 605524 | 601169 | 0.84 | 2.27E-06 | 4.08E-05 | 1.8 | 179 |
| mmu-miR-1839-5p | 4677 | 4335 | 4883 | 4813 | 2693 | 2812 | 2575 | 0.80 | 1.09E-05 | 1.62E-04 | 1.7 | 174 |

FC: fold change, % exp. = (Mean AE/Meanctr)*100

(miR-106b, miR-144, miR-16-1, miR-1981, miR-214, miR-28a, miR-335, miR-345 and miR-532) the difference in 5p- and 3p-arm expression between the two groups ranged between 10% and 33% ($P < 0.01$) as shown in Fig 5B.

## Prediction of the target genes of dysregulated host miRNAs

Potential targets of the 28 differentially expressed host miRNAs were predicted using the miR-Net tool which was based on miRTarBase v6.0, TarBase v6.0 and miRecords algorithms. In total 1645 target genes were identified for 25 miRNAs, while for the remaining three miRNAs (mmu-miR-148a-5p, mmu-miR-30a-3p and mmu-let-7d-3p), no target genes were found. Overall, from the 1645 targets, two clusters of unequal sizes were defined: a major group (1484/1645; 90%) includes target genes that are unique to one miRNA and a smaller set of 161 genes that are commonly controlled by two or more of the dysregulated miRNAs.

For the 17 down-regulated miRNAs, a set of 1426 target genes were found; almost 80% (1140/1426) of these miRNA-target interactions (MITs) were supported by experimental evidence. Among the 1426 target genes, 134 (10%) were shared by at least two miRNA molecules;

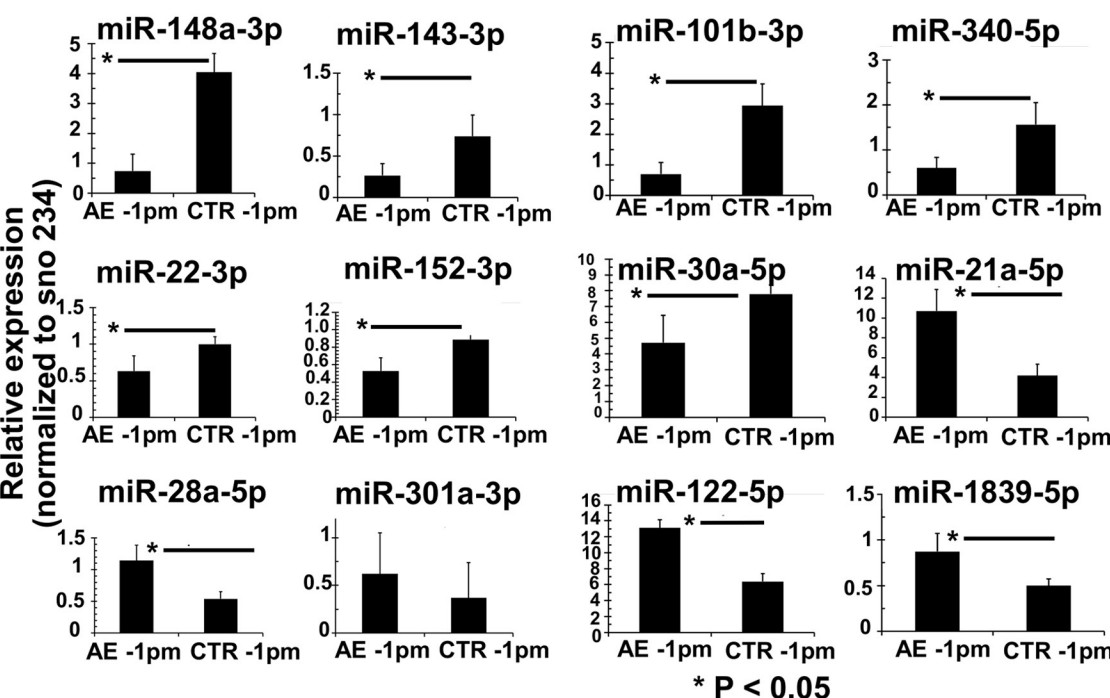

**Fig 4. miRNAs expression validation by stem-loop RT-qPCR.** Relative miRNA expression levels in livers of AE-infected and uninfected control mice. Normalization was done using *sno234* as the endogenous control. Bars represent median ± standard deviation. All stem-loop RT-qPCRs for each miR used biological replicates (n = 5 infected group and n = 4 uninfected group), with three technical replicates per experiment.

out of these 134 targets, 49% (66/134) were common between mmu-miR-15a-5p and mmu-miR-340-5p.

On the other hand, for the eight up-regulated miRNAs, 219 target genes were identified, and a smaller number of MITs (117/219; 50%) were experimentally validated. Among the 219 identified target genes, 89 were regulated only by mmu-miR-122-5p, with 71 MITs being experimentally validated. In total, twenty six genes were common between two microRNAs; mmu-miR-23a-3p and mmu-miR-23b-3p share 21 targets, representing 81% of genes targeted by two different up-regulated miRNAs.

The 1645 putative target genes covered a wide range of biological functions, notably those related to immunity, metabolism and epigenetic modifications such as DNA methylation and histone modification. Genes relevant to immunity included IL-1β targeted by mmu-miR-122-5p, SMAD3/4 transcription factors targeted by mmu-miR-27b-3p and mmu-miR-122-5p, calcium/calmodulin-dependent protein kinase II alpha (Camk2a) targeted by mmu-miR-340-5p, mmu-miR-148a-3p, mmu-miR-148b-3p and mmu-miR-152-3p, interferon regulatory factor (IRF) 7/8 targeted respectively by mmu-miR-122-5p and mmu-miR-22-3p, IL-17 receptor A (IL-17RA) targeted by mmu-miR-23a/b-3p, inducible T-cell costimulator (ICOS) targeted by mmu-miR-101a-3p and vascular cell adhesion molecule (V-CAM)- 1 target by mmu-miR-340-5p.

Two genes were involved in fatty acids (FAs) biosynthesis and activation; acyl-CoA synthetase long-chain family member 1 (ACSL1) and Fatty Acid Synthase (Fasn) targeted by mmu-miR-340-5p and mmu-miR-15a-5p.

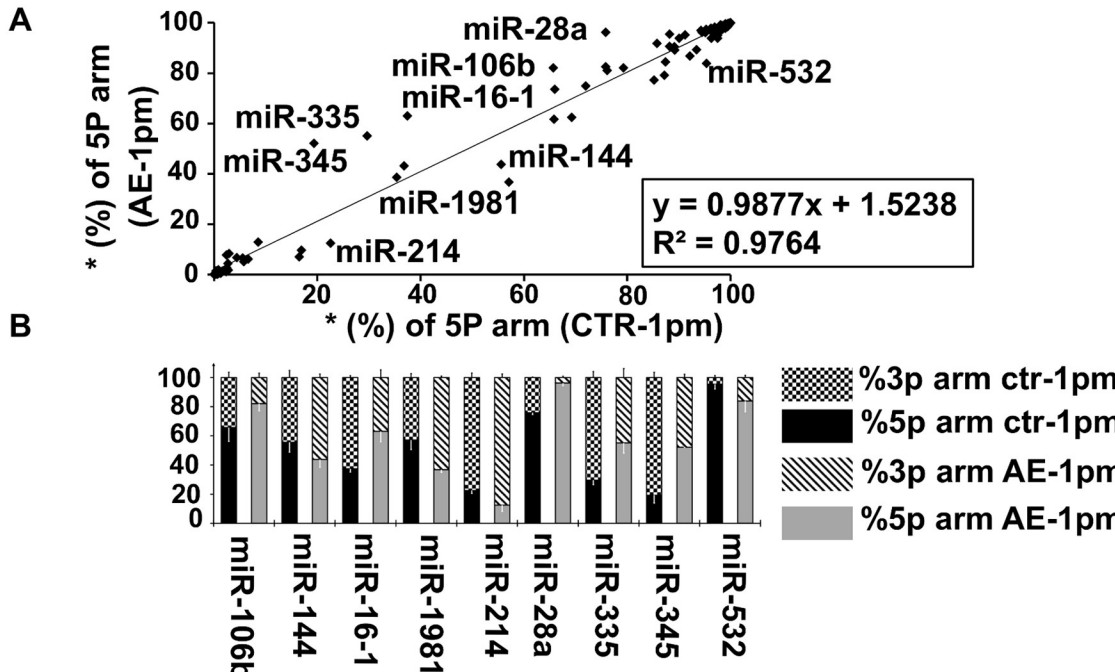

**Fig 5. Arm selection preference (5p and 3p) in hepatic pre-miRNAs: AE-infection versus no-infection (control).** (**A**) Linear regression analysis of the correlation in 5p-arm selection rate between *E. multilocularis*-infected (AE-1pm) and uninfected mice (ctr-1pm). The 5p-arm selection rate was determined for 127 miRNA pairs. (**B**) Nine pre-miRNAs exhibited a significant difference in 5p- and 3p-arm expression between *E. multilocularis*-infected (AE-1pm) and uninfected animals (ctr-1pm). For each of the nine miRNAs, the 100% stacked bars compare the 5p- and 3p-arm expression rate between AE-1pm and ctr-1pm mouse group. Black error bars represent standard deviation in percent of the 3p arm, and white bars represent standard deviation in percent of the 5p arm.

Relevant genes involved in epigenetic modifications included histone deacetylase (HDAC)-2/4/9 targeted by mmu-miR-340-5p, mmu-miR-22-3p and mmu-miR-340-5p, respectively, and DNA methyltransferases (Dnmt)-1, targeted by mmu-miR-148a/b-3p and mmu-miR-152-3p, respectively.

To elucidate regulatory relationships between up- or down- regulated miRNAs and the 1645 identified target genes, two interaction networks of miRNAs-mRNA targets were constructed and are shown in S2 and S3 Figs.

## Pathway enrichment analysis for dysregulated miRNAs

To get an overview on cellular pathways in which dysregulated miRNAs could be involved, the putative target genes were subjected to Reactome and KEGG for functional enrichment and pathway analysis. Analyses for down- and up-regulated miRNAs were made separately.

For down-regulated miRNAs: 56-Reactome- and 68 KEGG pathways ($P$-values < 0.01) were associated with up-regulated target genes. According to Fig 6A (Fig 6A), Reactome pathway enrichment analysis revealed that the main enriched biological processes targeted by down-regulated miRNAs were signaling pathways activated by growth factor receptors, namely vascular endothelial growth receptor 2 (VEGFR2), fibroblast growth factor receptor (FGFR), epidermal growth factor receptor (EGFR) and platelet-derived growth factor (PDGF). Twelve genes were shared between these four signaling pathways (S4 Table). A set of 25 genes

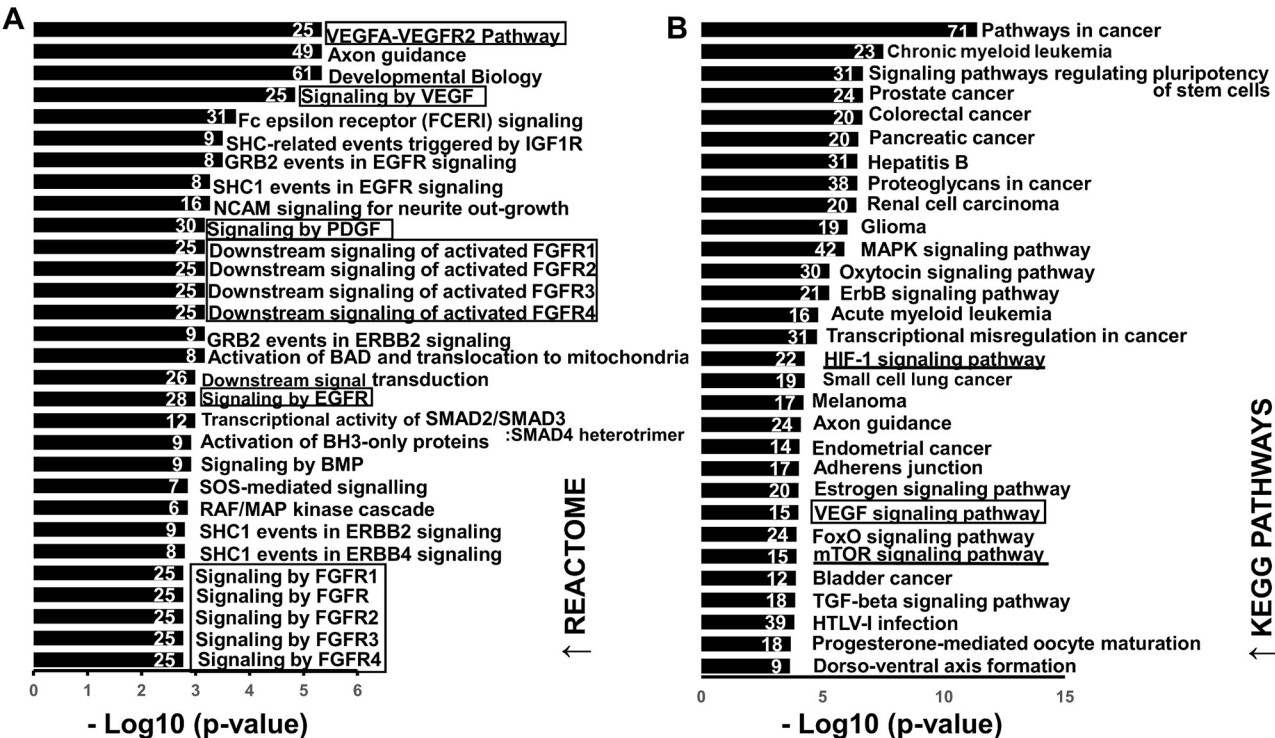

**Fig 6. Functional analysis of predicted targets of down- regulated miRNAs.** A and B: Top 30 overrepresented canonical pathways for up-regulated gene targets of down-regulated miRNAs according to Reactome and KEGG database, respectively. Pathways are ranked by score (-log10 (*P*-value). A higher score indicates that the pathway is more significantly associated with genes of interest. Numbers inside the end of each bar indicate the number of genes involved in each pathway.

involved in "VEGFA-VEGFR2" are highlighted in the network of down-regulated miRNAs-mRNA targets (S2 Fig).

The KEGG enrichment analysis revealed that target genes of down-regulated miRNAs were abundantly present in pathways that are involved in various cancers (Fig 6B).

Top 30 Reactome- and KEGG- -enriched pathways among up-regulated genes are shown in Fig 6 (Fig 6A and 6B). Genes associated to each Reactome and KEGG enriched pathway are listed in S4 Table.

As shown in Fig 7 (Fig 7), 6-Reactome and 23 KEGG pathways were significantly enriched in target genes of up-regulated miRNAs. Pathway analysis using Reactome database showed that heme biosynthesis was the most affected pathway with seven involved genes (Fig 7A) which are highlighted in the network of up-regulated miRNAs- mRNA targets (S3 Fig). Identified KEGG processes included mostly pathways associated with cancer (8 out the 23 KEGG pathways) (Fig 7B).

## Expression analysis of key pro-angiogenic and fatty acid synthesis – associated genes as targets of down-regulated miRNAs

Relative expression levels of five genes that were identified as targets of down-regulated miR-NAs and are involved in angiogenesis (*vegfa*, *mtor* and *hif1α*) and lipid metabolism (*fasn* and *acsl1*), were further assessed by RT-qPCR in livers from AE-infected and non-infected mice

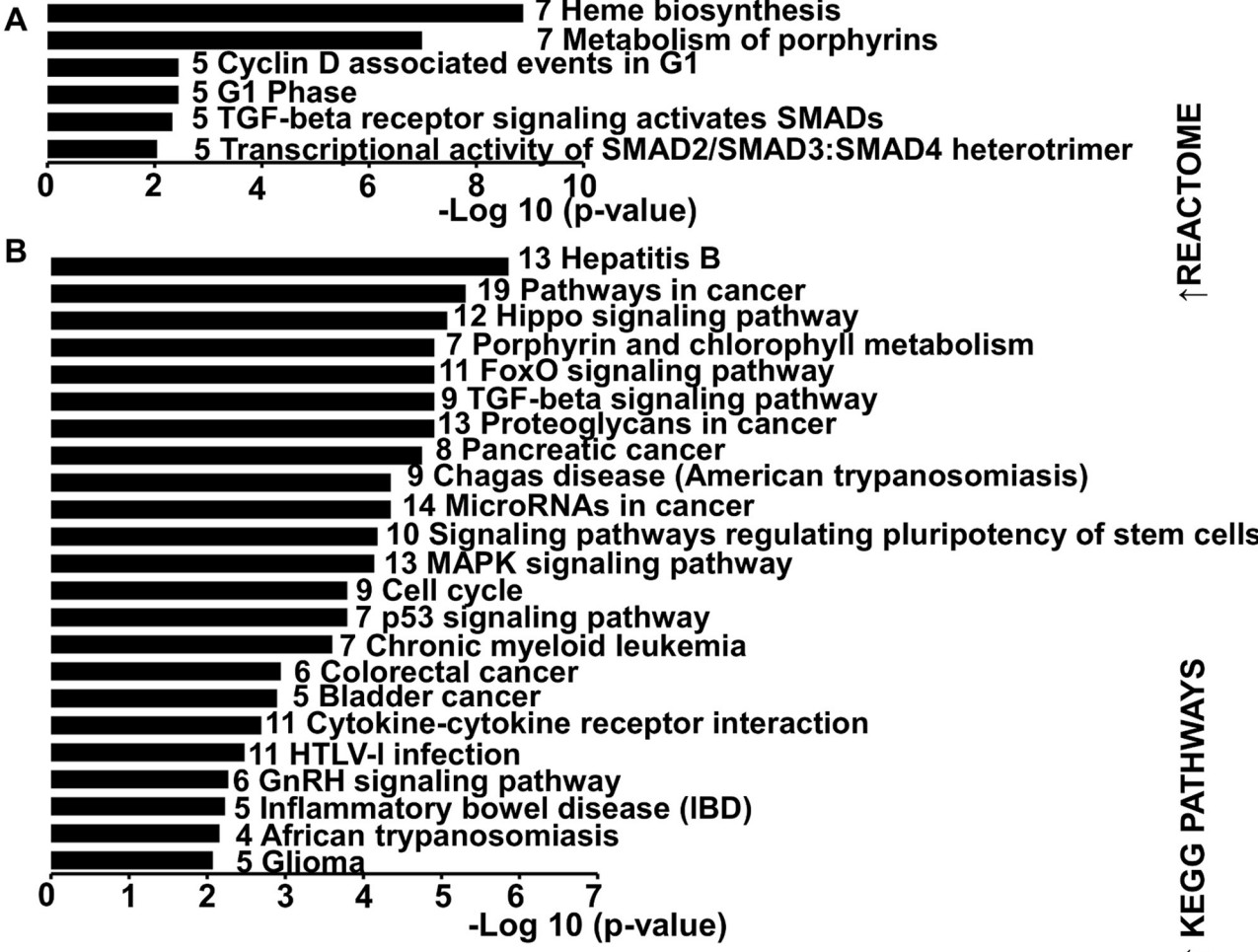

**Fig 7. Functional analysis of predicted targets of down- and up-regulated miRNAs.** A and B: Significant over-represented canonical pathways for down-regulated gene targets of up-regulated miRNAs according to Reactome and KEGG database, respectively. Pathways are ranked by score (-log10 (*P*-value)). A higher score indicates that the pathway is more significantly associated with genes of interest. Numbers inside the end of each bar indicate the number of genes involved in each pathway.

(S2 Table). As shown in Fig 8, all five genes exhibited significantly increased mRNA levels in livers from *E. multilocularis*-infected mice relative to the non-infected control group (Fig 8).

## Discussion

In the present study we undertook NGS-based miRNAs profiling in liver tissues of mice isolated at an early stage of primary AE. The cellular composition is heterogeneous, and includes hepatocytes, stellate cells, Kupffer cells and liver endothelial cells. In addition, hepatic AE is characterized by a periparasitic infiltration of immune cells [8]. Thus our differential expression analysis of hepatic miRNAs concerns all these cell types which might be present in the isolated liver tissue samples from infected and non-infected mice.

In total, we identified 28 miRNAs that were differentially expressed between infected and non-infected mice at 1 month post-infection. In general, alterations in the microRNA

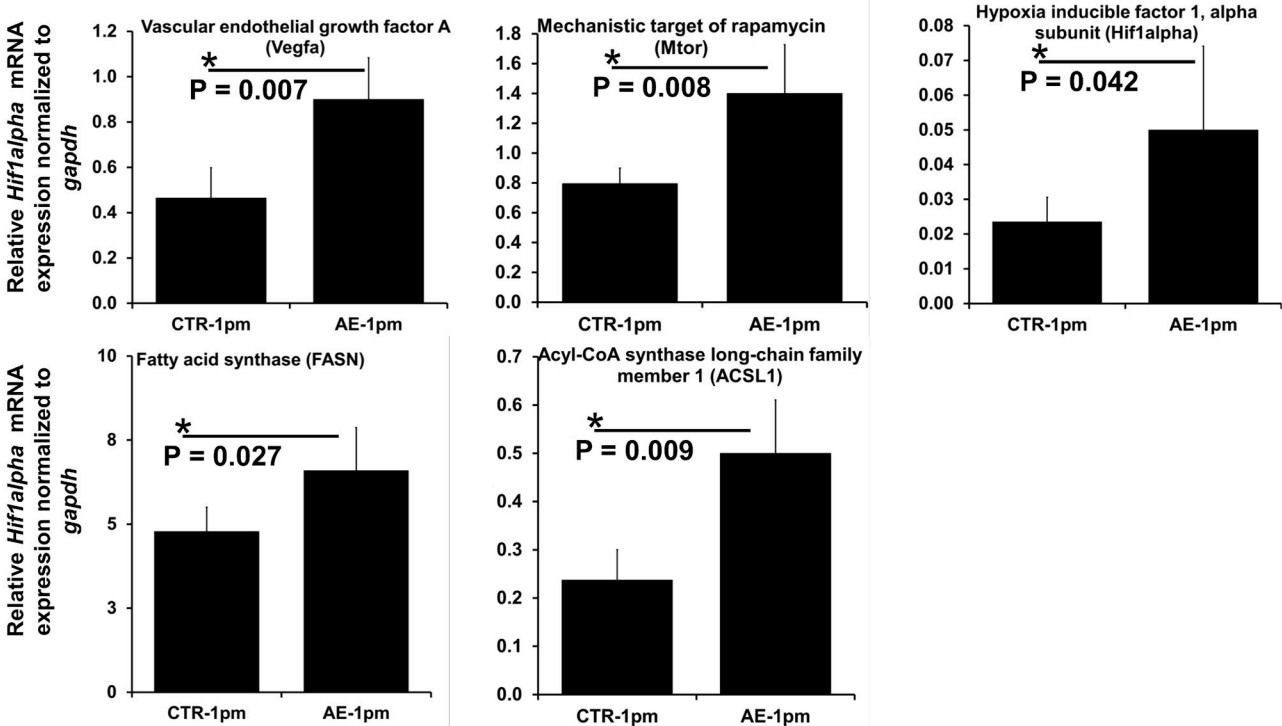

**Fig 8. Relative mRNA level of angiogenesis- and lipid metabolism-related genes.** Normalization was done using *gapdh* as endogenous reference. Bars represent median ± standard deviation. All qPCRs were carried-out using biological replicates (n = 5 mice from the infected group and n = 4 mice from uninfected group), with three technical replicates per sample.

transcriptome have been reported for a wide range of infectious- and non-infectious liver diseases [69].

Thirteen out of the 19 down-regulated miRNAs identified in our study were also down-regulated in experimental secondary AE in DBA/2 mice at three months post-infection [40,41]. However, none of the 9 differentially up-regulated miRNAs identified herein has been shown to exhibit altered expression levels in experimental secondary AE [40,41]. This unconformity is most likely due to differences between experiments such as the route of infection, mouse strain, and infection stage.

On the other hand, 19 out of the identified 28 dysregulated miRNAs were reported to be also affected in hepatocellular carcinoma (HCC) which is the most common type of primary liver cancer in adults [70–78]. Nonetheless, it remains to be shown whether changes in expression levels of the 19 miRNAs are common to various liver diseases, or whether they result from common induced cellular events/mechanisms that occur in liver cancerous cells as well in periparasitic liver tissue.

In our study, both miR-148a-5p, miR-148a-3p and miR-15a-5p were down-regulated and mRNA levels of the two lipogenic enzymes FASN and ACSL1 were higher in livers of *E. multilocularis*-infected mice when compared to controls. MiR-148a (5p and 3p) is directly involved in controlling cholesterol and triglyceride homeostasis [79–81]; down-regulation of miR miR-148a leads to an increased expression of genes related to lipogenesis and fatty acid uptake [82]. In the liver, overexpression of ACSL1 [83] results in increased proportion of oleic acid in

diacylglycerol (DAG) and phospholipids (PLs), and promotes synthesis of triglyceride (TG) from free fatty acids and its accumulation in hepatoma cells [84].

Fatty acid synthase (FASN) [85] is a primary target of miR-15a-5p [86–88], and its mRNA expression was found to be significantly up-regulated in mouse liver at early stage of AE infection [51]. However, one has to consider that a single mRNA can be targeted by many different miRNAs [89] and that regulation of gene expression can also mediated by post-translational modifications. Therefore the relative increase of *fasn* and *acsl1* expression levels can only be partially explained by the decrease of miR-148a and miR-15a-5p.

Another miRNA known to play important role in regulating cholesterol and fatty acid (FAs) metabolism is miR 122, a liver-specific miRNA [90–92]. In mice, inhibition of miR-122 by *in vivo* antisense targeting resulted in a decrease in hepatic FAs and cholesterol synthesis rates [93]. In our study, we observed a high expression of miR-122-5p in AE- infected mice. Since *E. multilocularis* is unable to synthesize FAs, cholesterol and other sterols *de novo*, [94] it is not surprising that AE might promote lipid synthesis in the liver. It was suggested that experimental AE in jirds could stimulate fatty acid biosynthesis [95], nevertheless further investigations are required to determine quantitative and qualitative effects of AE on host lipid metabolism and to define the exact role of miRNAs in this mechanism.

Since one miRNA may regulate many genes as its targets, it is also important to note that our results cannot be used to rule out the occurrence of other metabolic pathways which might be controlled by dysregulation of miR-148a, miR-15a-5p and miR-122-5p.

In macrophages, miR-148a-3p together with miR-30a-5p modulate inflammation by repressing NF-κB signaling and its respective pro-inflammatory consequences [96,97]. In this direction, downregulation of miR-148a-3p together with miR-30a-5p found in our study may suggest a role for these molecules in the gradual shift of Th1 to Th2 immunodominance associated with hepatic AE [7,8].

Interestingly, our pathway enrichment analysis of down-regulated miRNAs revealed a clear enrichment of growth factor-associated signaling pathways with VEGFA/VEGFR2 being ranked the first by statistical significance. In endothelial cells, binding of endothelial growth factor A (VEGFA) to VEGF receptor 2 (VEGFR2) results in the formation of new blood vessels from existing vessels [98,99]. Under hypoxic conditions, transcription of *vegfa* is promoted by hypoxia-inducible factor 1-alpha (HIF1-α) [100]. In our analysis, the HIF-1 signaling pathway was among significantly enriched KEGG pathways in the targeted genes of down-regulated microRNAs. In relation to miRNAs, both *vegfa* and *hif1-α* are experimentally validated targets of mmu-miR-15a-5p, mmu-miR-126a-5p and mmu-miR-101a/b-3p [101–105] (down-regulated miRNAs in this study). Herein, relative mRNA levels of *vegfa* and *hif1-α* were significantly higher in *E. multilocularis* infected liver tissues. In a previous study in Wistar rats, a higher protein level of HIF-1α was found in the actively multiplying infiltrative region of the AE liver lesions in comparison to the hepatic parenchyma [106].

The two miRNAs; mmu-miR-101a/b-3p mmu-miR-15a-5p have been reported to negatively regulate expression of the *mechanistic target of rapamycin* (*mTOR*) [107–109], which has recently emerged as a regulator linking inflammation to angiogenesis trough activation of the TNFα/IKKβ signaling pathway, which in turn lead to the production of extracellular matrix-degrading and remodeling enzymes [110–112]. In our experiment, we observed an increase in mRNA expression of *mtor* in infected mice, however further examination on the protein level is needed. Overall, although the presence of factors favoring formation of new blood vessels, precisely the tissue hypoxia caused by a continuous growth of metacestodes and the early host inflammatory reaction [113,114], the occurrence of angiogenesis during AE needs to be profoundly explored; it was recently demontrated that an angiographic vascularity takes place around liver lesions caused by *E. multilocularis* [115].

Target genes of miR-122 are mostly enriched in heme biosynthesis and porphyrin metabolism, this may be linked to the recently reported critical role of miR-122 for regulation of systemic iron metabolism [116]. To date there is growing evidence that miRNAs activate gene expression under certain conditions; nuclear miRNAs can activate gene transcription by targeting enhancers [117] and cytoplasmic miRNAs can function to post-transcriptionally stimulate gene expression [118]. Consequently, prediction analysis of cellular pathways that could be targeted for negative regulation by the up-regulated miRNAs must be approached cautiously and in a relative manner.

In infectious diseases, molecular mechanisms underlying the differential miRNA expression patterns remain unidentified. In addition, regulation of miRNAs expression is a complex and multilevel-process [119]. Expression of four genes involved in miRNAs biogenesis was found to be altered subsequently to liver infection with *E. multilocularis* [41], however this cannot explain the reported selectivity in regulating miRNAs expression. Further investigations are needed to clarify the contribution of parasites in changing expression and abundance of host miRNAs, particularly regarding the potential role of epigenetic mechanisms. In this respect, it has been reported that epigenetics, including DNA methylation and post-translational modifications of histones, regulates the expression of a notable number of miRNA genes [120].

In this study, we also examined arm selection preferences (5p or 3p) in miRNA pairs of normal and *E. multilocularis* infected liver tissues. In fact, it was widely reported that the double stranded pre-miRNAs produced only one mature functional miRNA at one arm, either 5p or 3p. This selection relied largely on the thermodynamic stability of the strands as a chief factor in determining which arm of the duplex will be incorporated in the RISC complex as functional miRNA. Recent evidence revealed that many pre-miRNAs may yield two mature products deriving from both arms (5p and 3p) with different selection rates [121–124]. Furthermore, it was shown that in response to a pathological condition e.g. cancer [125,126] and infection [127], the 5p- and 3p selection preference of some pre-miRNAs was altered. So this so-called "arm switching" phenomenon is now recognized as a miRNA post-transcriptional regulatory mechanism [128]. In our study, we found nine miRNAs showing significant differences in 5p- and 3p-arm selection preference between normal and infected liver tissue. Arm switching may be explained by the fact that the miR-5p and miR-3p resulting from the same pre-miRNA may act on different mRNA targets, thus they might be subjected to inverse regulation [129].

In conclusion, we demonstrated that primary murine AE significantly alters the hepatic miRNA transcriptome during the early stage (1 month post-infection). 28 miRNAs exhibited altered expression levels, with 19 miRNAs being down-regulated and 8 being up-regulated. Our analysis indicates that target genes of dysregulated miRNAs might be involved in angiogenesis and metabolic pathways, in particular lipid and heme biosynthesis. Future studies on transcriptome characterization of both mRNA and miRNA during more advanced AE-stages are urgently required with the prospective goal to first identify miRNAs signatures associated with disease progression and second to get a deep and direct insight into the cellular pathways which could be potentially targeted by dysregulated miRNAs.

## Supporting information

**S1 Table. Stem-loop RT-qPCR primers.**
(XLSX)

**S2 Table. qPCR primers.**
(XLSX)

**S3 Table. list of identified miRNAs in the control and infected samples.**
(XLSX)

**S4 Table. Reactome and KEGG pathways enriched for targets of the 19 down- and 9 up-regulated miRNAs.**
(XLSX)

**S1 Fig. Characterization of the miRNAs identified by NGS from AE-infected and control animals. (A)** Length distributions of total reads in the five small-RNA libraries. On the X-axis, reads < 10 nucleotides and reads > 25 nucleotides were discarded. The Y-axis depicts the read counts. The peak for the miRNA candidates (21 nucleotides) is centered. **(B)** The frequency of miRNAs that are expressed at the defined levels in each group, most of miRNAs are expressed with a read count less than 100.
(TIF)

**S2 Fig. Down-regulated microRNA-target network.** This network represents regulatory relationships between down-regulated miRNAs in alveolar echinococcosis and their target genes. Blue square: miRNAs and red dots: target genes. The 25 genes involved in VEGFA-VEGFR2 pathway are highlighted in green. The Network can be reproduced by entering the set of down-regulated miRNAs online in http://www.mirnet.ca/.
(TIF)

**S3 Fig. Up-regulated microRNA-target network.** In liver of *E. multilocularis*-infected mice, 9 miRNAs were significantly overexpressed as compared to uninfected controls. Target genes of seven up-regulated miRNAs were predicted. This network represents regulatory relationships between up-regulated miRNAs and their target genes. Two miRNAs (mmu-mir-1839-5p and mmu-mir-28a-5p) were not connected neither to the main tree nor to each other, thus they are not presented here. Twenty-six genes are common between two microRNAs (dark yellow circles). The seven genes involved in heme biosynthesis pathway are highlighted in green.
(TIF)

## Acknowledgments

The authors gratefully acknowledge Dr. Marcela Cucher for critical reading of the manuscript (Instituto de Investigaciones en Microbiología y Parasitología Médica, Buenos Aires, Argentina).

## Author Contributions

**Conceptualization:** Ghalia Boubaker, Markus Spiliotis.

**Data curation:** Ghalia Boubaker, Sebastian Strempel, Markus Spiliotis.

**Formal analysis:** Ghalia Boubaker, Sebastian Strempel, Markus Spiliotis.

**Funding acquisition:** Andrew Hemphill, Bruno Gottstein.

**Investigation:** Ghalia Boubaker, Norbert Müller, Junhua Wang, Markus Spiliotis.

**Methodology:** Ghalia Boubaker, Sebastian Strempel, Norbert Müller, Junhua Wang, Markus Spiliotis.

**Project administration:** Ghalia Boubaker, Bruno Gottstein.

**Resources:** Sebastian Strempel, Junhua Wang.

**Software:** Ghalia Boubaker, Sebastian Strempel.

**Supervision:** Bruno Gottstein, Markus Spiliotis.

**Validation:** Ghalia Boubaker, Sebastian Strempel.

**Writing – original draft:** Ghalia Boubaker.

**Writing – review & editing:** Andrew Hemphill, Bruno Gottstein, Markus Spiliotis.

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
