## [Decision Letter · Decision Letter 0]

2 Oct 2019

Dear Prof. Dr. Gottstein:

Thank you very much for submitting your manuscript "Regulation of Hepatic MicroRNAs in Response to Early Stage Echinococcus multilocularis Egg Infection in C57BL/6 mice" (#PNTD-D-19-01183) for review by PLOS Neglected Tropical Diseases. Your manuscript was fully evaluated at the editorial level and by independent peer reviewers. The reviewers appreciated the attention to an important problem, but raised some substantial concerns about the manuscript as it currently stands. These issues must be addressed before we would be willing to consider a revised version of your study. We cannot, of course, promise publication at that time.

We therefore ask you to modify the manuscript according to the review recommendations before we can consider your manuscript for acceptance. Your revisions should address the specific points made by each reviewer. 

When you are ready to resubmit, please be prepared to upload the following:

(1) A letter containing a detailed list of your responses to the review comments and a description of the changes you have made in the manuscript.

(2) Two versions of the manuscript: one with either highlights or tracked changes denoting where the text has been changed (uploaded as a "Revised Article with Changes Highlighted" file); the other a clean version (uploaded as the article file).

(3) If available, a striking still image (a new image if one is available or an existing one from within your manuscript). If your manuscript is accepted for publication, this image may be featured on our website. Images should ideally be high resolution, eye-catching, single panel images; where one is available, please use 'add file' at the time of resubmission and select 'striking image' as the file type. 

Please provide a short caption, including credits, uploaded as a separate "Other" file. If your image is from someone other than yourself, please ensure that the artist has read and agreed to the terms and conditions of the Creative Commons Attribution License at http://journals.plos.org/plosntds/s/content-license (NOTE: we cannot publish copyrighted images). 

(4) If applicable, we encourage you to add a list of accession numbers/ID numbers for genes and proteins mentioned in the text (these should be listed as a paragraph at the end of the manuscript). You can supply accession numbers for any database, so long as the database is publicly accessible and stable. Examples include LocusLink and SwissProt.

(5) To enhance the reproducibility of your results, we recommend that you deposit your laboratory protocols in protocols.io, where a protocol can be assigned its own identifier (DOI) such that it can be cited independently in the future. For instructions see http://journals.plos.org/plosntds/s/submission-guidelines#loc-methods

While revising your submission, please upload your figure files to the Preflight Analysis and Conversion Engine (PACE) digital diagnostic tool, https://pacev2.apexcovantage.com/ PACE helps ensure that figures meet PLOS requirements. To use PACE, you must first register as a user. Then, login and navigate to the UPLOAD tab, where you will find detailed instructions on how to use the tool. If you encounter any issues or have any questions when using PACE, please email us at figures@plos.org.

We hope to receive your revised manuscript by Dec 01 2019 11:59PM. If you anticipate any delay in its return, we ask that you let us know the expected resubmission date by replying to this email.

To submit a revision, go to https://www.editorialmanager.com/pntd/ and log in as an Author. You will see a menu item call Submission Needing Revision. You will find your submission record there. 

Sincerely,

Aaron R. Jex

Deputy Editor

Aaron Jex

Deputy Editor

Reviewer's Responses to Questions

**Key Review Criteria Required for Acceptance?**

**Methods**

-Are the objectives of the study clearly articulated with a clear testable hypothesis stated?

-Is the study design appropriate to address the stated objectives?

-Is the population clearly described and appropriate for the hypothesis being tested?

-Is the sample size sufficient to ensure adequate power to address the hypothesis being tested?

-Were correct statistical analysis used to support conclusions?

-Are there concerns about ethical or regulatory requirements being met?

Reviewer #1: The methods are appropriate.

There is one detail that I found in the methods that I do not understand:

Line 276: "We used cDNA samples prepared for the Stem-loop RT-qPCR for validating the set of 28 dysregulated mature miRNAs (see section above)." How could you use these cDNA samples for qPCR of target mRNAs? They were primed using specific stem-loop primers for each miRNA...

Reviewer #2: Some of the methods were not described in sufficient detail (see detailed comments below). Further evidence of how they determined levels of significance for miRNA differential transcription is required.

**Results**

-Does the analysis presented match the analysis plan?

-Are the results clearly and completely presented?

-Are the figures (Tables, Images) of sufficient quality for clarity?

Reviewer #1: Yes. It would be great to have as supplementary information a complete table of miRNA expression in each sample, including also miRNAs that did not show any significant change (i.e. a full version of table 4). I would also recommend sharing the raw RNAseq data (as far as I could see, these have not been made available in a public database).

Reviewer #2: Results clearly match the aims of the study. The results are presented in sufficient detail. The link between their results in fatty acid biosynthesis were not very clear. Perhaps more information is required (pathway genes listed? Synonyms made clear for a broad audience?)There are a lot of tables and figures presented in the main body of the manuscript. Some consolidation of figures and moving quality control results to the supplementary material is recommended. Additional comments about the results are listed below.

**Conclusions**

-Are the conclusions supported by the data presented?

-Are the limitations of analysis clearly described?

-Do the authors discuss how these data can be helpful to advance our understanding of the topic under study?

-Is public health relevance addressed?

Reviewer #1: - The main issue that I find lacking in the discussion is that there is no hypothesis regarding how AE changes miRNA expression in the surrounding tissues. In particular, an aspect that is not considered is that part of the differences could be related not only to differential gene expression of constituent hepatic cells, but also from the different cell composition that results from the inflammatory immune response (accumulation of macrophages, lymphocytes, etc, in the surrounding tissues).

 - I am not convinced about some of the comparisons between AE and HCC - specifically, in lines 444 to 448, you compare miRNA changes between both conditions. It is true that the parasite grows like a tumour, and may affect the surrounding tissues in some similar ways. However, here you are comparing miRNA expression of tumour cells for HCC vs. miRNa expression in the surrounding tissues for AE.

- There are several enriched pathways in figure 7 that are related to each other and probably share constituent genes. It seems likely to me that many of the genes that result in VEGFA-VEGFR2 pathway enrichment are also resulting in the enrichment of PDGF, EGFR, FGFR pathways (many of the genes indicated in Supplementary Figure 1 are largely shared between these pathways, and as far as I can see, only VEGFA is specific). In my opinion, this should be part of the description and discussion of these results, as it tones down the specific importance of VEGF. It would be good if the genes involved in pathway enrichment for each category were provided as supplementary information. 

- Even though I find this study superior in execution and design (as it involves samples from primary echinococcosis and greater statistical power), it would be worthwhile to include a more detailed comparison to the miRNAs found to be dysregulated by AE by Jin et al 2017 (cited by the authors) as some coincide between both studies. 

- Although typically mRNA target levels decrease in the presence of the targeting miRNA, the precisse relationship between miRNA and target mRNA levels can be quite variable, as most regulation occurs at a translational level. Therefore, you could mention that it is likely that the true effect on gene expression is subestimated in the qPCR experiments.

Reviewer #2: The discussion is quite long and the authors have made a number of conclusions from this data set. I would recommend they be more cautious considering the small changes in transcription that were observed (particularly the mRNA validation).

**Editorial and Data Presentation Modifications?**

Reviewer #1: Line 24 and Line 66: Explaining the abbreviation "E. multilocularis" is unnecesary (it is standard to abbreviate the genus to its inital letter after the first time that the species is named).

Line 66: cyclophyllid -> cyclophyllidean

Line 100: the number of true miRNA loci is still very controversial - see for example doi: 10.1093/nar/gkz097. Perhaps mention a range of possible miRNA loci instead based on the literature.

Line 194: this should probably be "libraries from five mice"

Line 199: "intended" -> "expected". Is the 141 bp peak the result of the addition of the 21 bp miRNA cDNA plus the adapters?

Line 238: did you use the same primers for sno234 as in ref. 62?

Line 243: change "5 min_25ºC" to "5 min at 25 ºC", etc.

Line 250: add the missing parenthesis

Line 270: "Precisely" -> "Specifically"

Line 302: "compromised" -> "comprised"

Line 308: you generally used miRNA as singular and plural throughout the text, but here you used miRNAs as the plural

Line 310: "whose log2FC" -> "whose fold change"

Line 312: "FC ≤ 0.6" should probably be "FC ≤ 0.67"

Line 334: "they" -> "that"

Line 344 to 348: it would be more clear if you reversed the order of these sentences (and would result in mentioning figure 6a before figure 6b).

Line 352: "identified for the 25 miRNAs" -> "identified for 25 miRNAs"

Line 366: "alone mmu-miR-23a-3p and mmu-miR-23b-3p share 21 targets, 81% of common genes." -> I would recommend changing this to mmu-miR-23a-3p and mmu-miR-23b-3p share 21 targets, representing 81% of genes targeted by two different up-regulated miRNAs."

Line 369: "Immune relevant" -> perhaps change to "Genes relevant to immunity included..."

Line 369-378: in some cases, the same mRNA is predicted to be targeted by different miRNAs that are up and down-regulated, so the expected effect is not clear. This could be mentioned here.

Line 377: ACSL1 is not directly involved in FA biosynthesis, but in FA activation for complex lipid synthesis and catabolism.

Line 383 is not very well connected to the rest of the text.

Line 397: It could be worthwhile to mention that all mRNA related to heme biosynthesis are targeted by miR-122, an hepatic miRNA with known roles in iron homeostasis (doi: 10.1172/JCI44883)

Line 401: what do you mean here by "most significant"? They were not specifically among the enriched categories shown in Fig 7.

Line 443: "metastatis" -> "metastasis"

Line 469: actually, FASN has several domains and catalyzes all steps of FA synthesis from acetylCoA and malonylCoA.

Line 518: I believe this line should be toned down; the results suggest that this phenomenon is partially mediated via downregulation of specific miRNAs

Line 563: "evaluate" -> "to evaluate"

Line 571: "Marcella" -> "Marcela"

- In my opinion, Figure 1 could be transferred to the Supplementary Information.

Reviewer #2: Grammatical and unclear text

Lines 54 to 56. Sentence unclear, please check grammar

Lines 84. Sentence is unclear. Please check for grammatical errors

Lines 100 to 101. Sentence is unclear. Perhaps it needs to be clear what a gene is first (coding and non-coding)? 

Lines 141 to 142. Unclear if you are referring to the miRNA or the mRNA that they target.

Line 184. Can you include the company and product information for the DNase I?

Lines 201 to 202. Methods is unclear here.

Lines 202 to 204. This statement is unclear. How long were the reads sequenced? is 50 Mio 50 million. What is past filter?

Lines 218 to 219. Not clear what they mean by valid here? Please revise and make it clearer.

More detail is required for the headers/description of Tables. Table 3 is difficult to understand with the information provided in the headers

Lines 403 to 407. This reads like methods to me

Lines 56. You mention pathways but have not introduced the pathways you are referring to.

Line 407. Levels... typo

Lines 423 to 424. Sentence is unclear

Body of the manuscript

Beginning of the abstract reads more like a summary of the materials and methods. I don't think the methods needs to be included in the abstract with this much detail. Summary introduces hypotheses generated from the results that were not mentioned in the abstract. Abstract needs to be revised to include more results and less materials and methods.

Introduction is very long. Some information is not required. The role of miRNA is very well established now. Perhaps the role of miRNA in parasite infections is all that needs to be introduced (i.e. human miRNA and their role is already well established and needs no introduction here).

Lines 157 to 165. I assume this method is well established and has been published previously? Perhaps there is no need to detail the methods if it has already been published?

Figures 1A could be presented as one graph. Each line would be a different library. Further consolidation of the figures in this paper would allow them to be printed larger and improve the readability. This figure could also be removed to Supplementary files.

Grey labelling of the different miRNA in Fig 2B is not described in the Figure legend. What is the significant of the different colours? Common and unique to the different groups? Some additional description in the Figure legend

would be required.

Fig 6B description is before 6A. Panels need to be swapped on Fig 6.

Discussion section is quite long. Conclusion paragraphs could be significantly reduced. For example, there is quite a bit of talk about biomarkers and this is not required here, simply mentioned at the end of the last sentence perhaps?

Tables 1 and 2 could be moved to supplementary tables. Table 3 requires more detail to improve understanding of each column

**Summary and General Comments**

Reviewer #1: This is an interesting, well executed and clearly written article in which the authors compare the miRNA transcriptome of hepatic tissues in control conditions and during early AE infection. This is a primary, hepatic infection, which is a good model for the natural infection in rodents. The authors strenghten their RNAseq results by confirming a selected number of significantly regulated miRNAs by qPCR, and they also analyze the levels of some predicted miRNA targets by qPCR. 

I only have some observations and recommendations regarding the discussion of the results, and a question regarding the methods, which I already included in the previous sections. I also provided some editorial comments.

Reviewer #2: It is well established that flatworm parasites (including E. multilocularis) have the capacity to regulate their host immune response. The molecular mechanisms they use to do this are still poorly understood. Understanding changes in host mRNA and miRNA transcription in response to the intial infection will provide new insights into this exciting research area. The authors describe changes in miRNA transcription in mice livers one month after being infected with E. multilocularis (early in the infection). Results were validated for select miRNA and mRNA using qPCR. Significantly differently regulated miRNA pointed to changes in the regulation of target mRNA and pathways they are associated with. Specifically, angiogenesis, axon guidance and ??fatty acid biosynthesis?? The authors use their results to support the role of known pathways associated with parasite infection as well as proposing novel changes in molecular signalling in the liver in response to parasite infection. I observed a few limitations with this study (sample size for DESEQ2 and fold changes observed) and care should be taken from drawing too many conclusions from their results. My main concern was the lack of detail in the materials and methods section that made it difficult to assess why they chose less stringent fold-change cutoffs (1.5 fold change is quite low; was multiple testing correction used?) for changes in transcription as well as only looking at the miRNA with >1000 read counts (only a small fraction of their data). Importantly, there was no evidence that they have submitted their raw small RNA sequence reads to a public archive (e.g. NCBI SRA). This is required. I outline my concerns in detail below. 

Major comments

When annotating the miRNA using BLAST: Would you not be looking for a perfect match to these nucleotide sequences? Please justify this two step process of annotating putative miRNA. Conducting BLAST searches against miRBASE and Rfam may not be the most reliable method of annotating your miRNA. For example, Rfam would recommend using Infernal (HMM-based searching) to annotate small RNA using the Rfam database.

Lines 219 to 221. The link from small RNA clustering, annotation to mapping is unclear. When mapping to the genome, how were the locations of your miRNA determined. Usually the premature miRNA would be used for positioning the miRNA within the genome. In this case, were you mapping to the mature, the star or the mature, star and hinge all at the same time? More information is required here.

Line 225. If this is the case, then how did you annotate miRNA that were not a match to miRNA encoded in the reference genome? (see Lines 217 to 218)

Lines 226 to 227. Were the distributions of miRNA transcription normally distributed? Can often be tricky when performing pairwise comparisons tests using skewed data (often observed with miRNA data). Also, what settings were used?; fold change, and multiple testing correction q value? I assume multiple test correction was performed as part of the DESeq2 pipeline. Overall, more information should be included about the steps taken to determine the differentially transcribed miRNA. Currently it is too difficult to work out what has been done. 

Could the authors also comment on the use of the DESEQ2 pipeline when your number of replicates is 2 in the control group?

Lines 229 to 230. Not sure what you mean about the 2D here. Do you mean that you chose to only display 2 dimensions? Presumably the 2 dimensions that accounted for the most variation?

Line 265. What was used as a cutoff for measuring significance? Pvalue < 0.05/0.01?

Line 277. Unclear which 28 mature miRNA are being referred to here. There are less than 28 primer sets listed in Table 1. 

Lines 293 to 294. So most miRNA had mapped reads aligned at a depth of 10 to 100? Is this quite low or normal for mouse studies? Unclear why you then chose to use a depth cutoff of 1000 (see Line 312). You are removing most of your data. Particularly as it was in combination with such a low fold change coverage (1.5 fold change)? Further justification for your methods is required.

In Figure 2A, you refer to 124 AE-specific miRNAs with read counts < 100. Do you mean > 100?

Methods for Figure 3B are not included in the materials and methods section. Unclear what values are used in the legend on Fig 3B. More information required.

Lines 312 to 317. How can let 7 miRNA be up and down regulated? star and mature miRNA? More detail on let 7 genes may need to be described. Is this an example of arm swapping?

Fig 6B. Unclear what the error bars represent in this panel. It is a stacked bar plot which equals 100%

Lines 383 Production of this result is not described in the materials and methods section. I suggest they be deleted. There is also no description of this result.

Lines 401 to 402. Here you mention that fatty acid synthesis was a significantly afffected pathway but earlier you state that it is angiogenesis and axon guidance. I couldn't see the link between axon guidance and fatty acid synthesis. More information is required here.

Lines 424 to 426. Without comparing miRNA profiles between human AE and other liver diseases it is difficult to see how you this statement is supported by your research.

Lines 516 to 518. Discussion concludes that there is strong evidence. Levels of transcriptional change are quite low for many of the genes you tested. I would be cautious in drawing too many conclusions from the data. At least, change the wording to make it clear that you are developing hypotheses that require further testing.

PLOS authors have the option to publish the peer review history of their article (what does this mean?). If published, this will include your full peer review and any attached files.

Reviewer #1: No

Reviewer #2: No

---

## [Decision Letter · Decision Letter 1]

18 Feb 2020

Dear Prof. Dr. Gottstein,

Thank you very much for submitting your manuscript "Response letter: PNTD-D-19-01183 [EMID:2740a10eb62af13a] “Regulation of Hepatic MicroRNAs in Response to Early Stage Echinococcus multilocularis Egg Infection in C57BL/6 mice”." for consideration at PLOS Neglected Tropical Diseases. As with all papers reviewed by the journal, your manuscript was reviewed by members of the editorial board and by several independent reviewers. The reviewers appreciated the attention to an important topic. Based on the reviews, we are likely to accept this manuscript for publication, providing that you modify the manuscript according to the review recommendations. 

The authors have satisfactorily addressed the reviewers' concerns, only minor editorial editions need to be addressed following reviewers' suggestions.

Sincerely,

Gabriel Rinaldi

Associate Editor

Aaron Jex

Deputy Editor

The authors have satisfactorily addressed the reviewers' concerns, only minor editorial editions need to be addressed following reviewers' suggestions.

Reviewer's Responses to Questions

**Key Review Criteria Required for Acceptance?**

**Methods**

-Are the objectives of the study clearly articulated with a clear testable hypothesis stated?

-Is the study design appropriate to address the stated objectives?

-Is the population clearly described and appropriate for the hypothesis being tested?

-Is the sample size sufficient to ensure adequate power to address the hypothesis being tested?

-Were correct statistical analysis used to support conclusions?

-Are there concerns about ethical or regulatory requirements being met?

Reviewer #1: (No Response)

Reviewer #2: Satisfied with most changes made in R1. Response to my comments on the experimental design and use of statistics was not complete, but the design is robust enough for the revised conclusions in R1

**Results**

-Does the analysis presented match the analysis plan?

-Are the results clearly and completely presented?

-Are the figures (Tables, Images) of sufficient quality for clarity?

Reviewer #1: (No Response)

Reviewer #2: Satisfied with the changes made in R1

**Conclusions**

-Are the conclusions supported by the data presented?

-Are the limitations of analysis clearly described?

-Do the authors discuss how these data can be helpful to advance our understanding of the topic under study?

-Is public health relevance addressed?

Reviewer #1: (No Response)

Reviewer #2: Satisfied with the changes made in R1

**Editorial and Data Presentation Modifications?**

Reviewer #1: A few additional editorial modifications:

Line 247 - some corrections were missing 

Line 308-310 - you mention first 45 ctr-1pm specific miRNAs, but then 24 ctr-1pm specific miRNAs

Line 375 and line 541 - you mention 8 up-regulated miRNAs, but they are actually 9, as mentioned elsewhere

Line 542 -"cautiously indictaes" - I would remove "cautiously";change indictaes for indicates

S Fig 1 - the x axis says "lenght" instead of "length"

Reviewer #2: Satisfied with the changes made in R1

**Summary and General Comments**

Reviewer #1: The authors have satisfactorily modified the manuscript and replied to the reviewers.

I have only included a few additional editorial comments above.

The only additional thing that I have not found in the manuscript is where the original sequencing data was deposited.

Reviewer #2: Thank you for addressing the majority of my comments

PLOS authors have the option to publish the peer review history of their article (what does this mean?). If published, this will include your full peer review and any attached files.

Reviewer #1: No

Reviewer #2: No
---

## [Editor Report · Decision Letter 2]

5 Mar 2020

Dear Prof. Dr. Gottstein,

We are pleased to inform you that your manuscript 'Response letter: PNTD-D-19-01183 [EMID:2740a10eb62af13a] “Regulation of Hepatic MicroRNAs in Response to Early Stage Echinococcus multilocularis Egg Infection in C57BL/6 mice”.' has been provisionally accepted for publication in PLOS Neglected Tropical Diseases.

Best regards,

Gabriel Rinaldi

Associate Editor

Aaron Jex

Deputy Editor

---

## [Editor Report · Acceptance letter]

6 May 2020

Dear Prof. Dr. Gottstein,

We are delighted to inform you that your manuscript, "Regulation of Hepatic MicroRNAs in Response to Early Stage *Echinococcus multilocularis* Egg Infection in C57BL/6 mice," has been formally accepted for publication in PLOS Neglected Tropical Diseases.

Best regards,

Serap Aksoy

Editor-in-Chief

Shaden Kamhawi

Editor-in-Chief
